# Ecological Risk Assessment and Prediction Based on Scale Optimization—A Case Study of Nanning, a Landscape Garden City in China

**Jianjun Chen** [1,2,*] 🆔**, Yanping Yang** [1]**, Zihao Feng** [1]**, Renjie Huang** [1]**, Guoqing Zhou** [1,2] 🆔**, Haotian You** [1,2] **and Xiaowen Han** [1,2]

1   College of Geomatics and Geoinformation, Guilin University of Technology, Guilin 541004, China
2   Guangxi Key Laboratory of Spatial Information and Geomatics, Guilin University of Technology, Guilin 541004, China
*   Correspondence: chenjj@glut.edu.cn

**Abstract:** Analysis and prediction of urban ecological risk are crucial means for resolving the dichotomy between ecological preservation and economic development, thereby enhancing regional ecological security and fostering sustainable development. This study uses Nanning, a Chinese landscape garden city, as an example. Based on spatial granularity and extent perspectives, using 30 m land use data, the optimal scale for an ecological risk assessment (ERA) and prediction is confirmed. This study also explores the patterns of spatial and temporal changes in ecological risk in Nanning on the optimal scale. At the same time, the Patch-generating Land Use Simulation model is used to predict Nanning's ecological risk in 2036 under two scenarios and to propose ecological conservation recommendations in light of the study results. The study results show that: a spatial granularity of 120 m and a spatial extent of 7 km are the best scales for ERA and prediction in Nanning. Although the spatial distribution of ecological risk levels is obviously different, the overall ecological risk is relatively low, and under the scenario of ecological protection in 2036, the area of high ecological risk in Nanning is small. The results can provide theoretical support for ERA and the prediction of landscape cities and ecological civilization construction.

**Keywords:** ecological risk; scale effects; scenario simulation; ecological conservation; Nanning

## 1. Introduction

A large number of ecological and environmental problems have emerged, including reduced forest coverage rate [1], degraded ecosystems [2], increased soil erosion [3], and loss of biodiversity [4], following rapid economic and social development, urbanization, the increasingly prominent contradiction between man and land, and the forced change of the structure and pattern of many ecological lands [5]. Human beings have brought about many negative ecological and environmental impacts while promoting socio-economic development [6]. On the other hand, nowadays, human power has been regarded as the primary source of power to improve environmental systems and can be used to reduce ecological risks, improve regional natural environments, and adjust the internal structure of ecosystems with urban planning [7], energy conservation and emission reduction [8], and carbon peaking and neutrality [9]. Thus, in order to manage urban ecological risk, plan for and restore the environment, and encourage the sustainable development of cities, people, and nature, it is indispensable to model and predict the changes in ecological risks under various scenarios and analyze the evolution characteristics of ecological risks in rapidly developing cities.

Landscape ERA, an extension of landscape ecology, is a method for monitoring and evaluating the negative impacts of human activities and the natural environment on ecosystem structure and function [10,11]. The landscape ecological risk values determine

the influence of landscape spatial patterns on ecological risk processes and functions, with typical spatial heterogeneity and scale effects [12,13]. The scale effect is a fundamental feature of spatial heterogeneity, and the degree of heterogeneity and dynamic processes of landscape patterns differ significantly at various scales [14], with different ecological risk evaluation results. Therefore, a suitable research scale is a prerequisite for ERA and can improve its accuracy and reliability.

At present, remote sensing data is the primary data source for environmental monitoring [15] and management [16], ecosystem protection [17], landscape patterns [18], and so on. In particular, the scale effect of ecological risk is closely related to the selection of remote sensing data. Scale effects are usually divided into spatial granularity and extent, and the appropriate scale for different study areas is not universal [19,20]. For the appropriate spatial granularity selection, current research has focused on the response of the landscape pattern index to its changes [21,22], the selection of granularity analysis methods, and the construction of models [23,24]. The landscape pattern index has been widely used in landscape-scale spatial analysis because it is highly condensed with information related to the landscape pattern and can better describe the structural composition and spatial configuration of different landscape elements [23]. In the analysis of the spatial granularity effect of landscape pattern indices, scholars usually adopt the resampling method [25] to analyze the scale effect of the study area and use the area loss evaluation [26] and the inflection point identification methods of the landscape pattern response curve [27] to determine the suitable granularity of the study area landscape. Although current studies have explored the effects of suitable spatial grain size on landscape pattern changes from a landscape pattern perspective, the impact of scale effects on landscape ecological risk has been ignored.

There are two methods for assessing landscape ecological risk, which are based on risk sources and sinks [28] and landscape patterns [29]. The method based on landscape patterns usually evaluates the ecological risk of the area directly from the perspective of the spatial pattern [30], involving the rational selection of ERA units and the direct reflection of spatial extent in the landscape ERA. Depending on the evaluation area and the study's purpose, assessment units can be delineated according to natural geographical units or artificially divided, mainly by directly taking watersheds [31], administrative districts [32], and nature reserves [33] as the ERA units, thus ensuring the structure's integrity and natural elements processes but, to a certain extent, ignoring spatial heterogeneity. The artificial division method mainly uses the risk cell or grid as the evaluation unit of ecological risk [34]. Risk cells are the smallest units in the evaluation, and too small of a division extent will destroy or even change the internal structure's integrity and the landscape's function. At the same time, too large of a division scale will lead to a loss in information about the landscape patches and will not fully or accurately reflect the actual situation inside the landscape. Therefore, a suitable spatial extent is a basis for dividing risky plots into those that can genuinely and effectively carry out an ERA.

ERA aims to carry out risk prevention and ecological protection based on its results. Simulating ecological risks under different scenarios is beneficial for comparing and studying the effectiveness of different protection measures. The simulation and prediction of ecological risk are usually based on land use (LU). At present, the primary land use simulation models are the Markov [35], the CA [36], the CLUE-S [37], and the Patch-generating Land Use Simulation (PLUS) models [38]. Among these models, the Markov model is widely used but has limitations in simulating spatial changes in LU. The CA model has mighty spatial computing powers, but its different conversion rules will lead to different LU simulation results. The PLUS model can respond to the drivers of LU change and their contribution rates and has better LU simulation accuracy. It has been widely used in LU simulation [33], carbon stock prediction [39], ecosystem service value models [40], etc.

Nanning is the political and economic center of Guangxi, China, and is known as the "Green City of China", one of the first "National Ecological Garden Cities" in China, and a "Beautiful Mountain City" for three consecutive years. The urbanization rate in

Nanning was estimated at 68.91% in 2020, so coordinating economic development with ecological protection and optimizing the spatial pattern of cities and towns is both the focus of and difficulty for Nanning's future sustainable development. Currently, there is little research focused on evaluating and understanding the ecological risks in Nanning and the simulation of future scenarios. There is an urgent need to analyze and predict the phenomenon of changes in ecological risks in the region and to propose appropriate ecological protection strategies. Using LU data, this study determines the optimal scale for ERA based on the granularity response curve of the landscape pattern index and the relevant results of the semi-variance function of ecological risk in the landscape. This study uses the ERA model to analyze the spatial and temporal variation characteristics of ecological risk in Nanning at the optimal scale and then simulated and predicted the changes in LU patterns and ecological risks for the year 2036. Our study explored the scale effects of ecological risks from the perspectives of spatial granularity and spatial amplitude, which improved the single perspective of previous studies on scale effects and improved the accuracy of ERA to a certain extent. At the same time, based on the research results, suggestions for the ecological protection of Nanning are put forward to ensure the rational planning and layout of landscape garden cities under the rapid development of cities and towns.

## 2. Materials and Methods

### 2.1. Study Area

Nanning is located in the central south of the Guangxi Zhuang Autonomous Region of China, at 107°45′–108°51′E, 22°13′–23°32′N. As of 2021, Nanning has seven districts (Qingxiu, Xingning, Jiangnan, Xixiangtang, Liangqing, Yongning, and Wuming), four counties (Bingyang, Shanglin, Longan, and Masan), and one county-level city (Hengzhou), with a total area of 22,112 km². It is the core city of the Beibu Gulf Economic Zone in Guangxi, with Guangdong, Hong Kong, Macao, and Qiong to the east, the Indian Peninsula to the west, Southeast Asia to the west, and the Great Southwest to the back [41] (Figure 1).

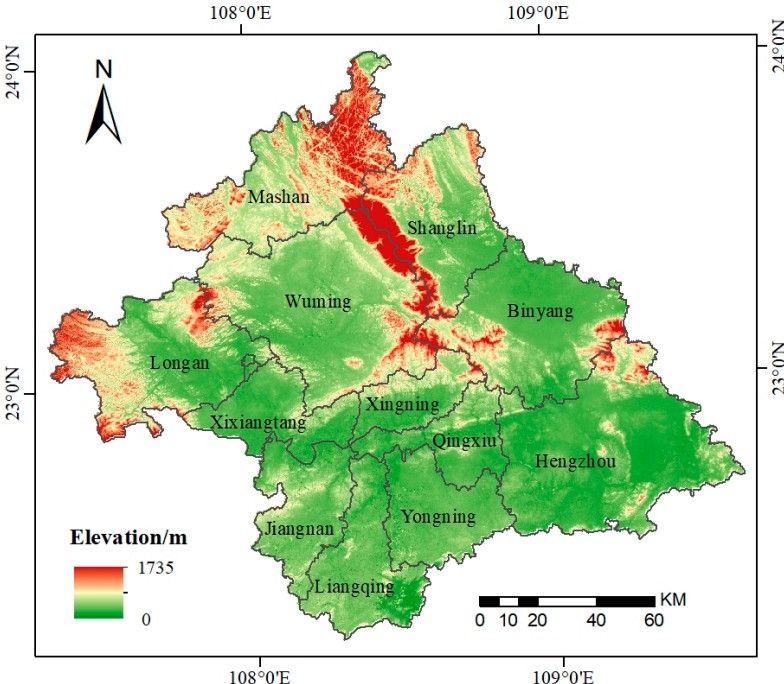

**Figure 1.** Elevation of Nanning.

There are five main types of landforms in Nanning: flat land, low mountains, rocky hills, hills, and terraces, of which flat land is the largest, accounting for 57.78% of the city's area. Located south of the Tropic of Cancer, Nanning has a subtropical monsoon climate

with abundant rainfall and sunshine with an average annual temperature of 21.6 °C and rainfall of 1304.2 mm [42]. The city has a well-developed river system and abundant water resources, with the main rivers belonging to the Xijiang River System of the Pearl River Basin, the largest river in Guangxi. Nanning is rich in flora and fauna, with 21 existing natural reserves, and is a pivotal city connecting the economic spheres of South China and Southwest China, as well as a regional international city in China facing the ASEAN countries. By the end of 2020, the city had a registered population of 7,912,800 and an urban population of 4,093,200 with a gross domestic product (GDP) of RMB 472,634 million, mainly in the tertiary and secondary industries.

### 2.2. Data Source and Pre-Processing

#### 2.2.1. Remote Sensing Product Data

The LU data used in this study were obtained from the Center for Resources and Environmental Sciences and Data of the Chinese Academy of Sciences (https://www.resdc.cn/ (accessed on 14 January 2021)). The data were produced by the Chinese Academy of Sciences and other institutions based on the Landsat remote sensing image data of the United States using artificial visual interpretation. According to the land resources and their utilization attributes, the land was divided into six types of LU, namely arable land, woodland, grassland, water, construction land, and unused land, with a spatial resolution of 30 m. According to the research needs, this study selected the LU data of 2000, 2010, and 2018 and trimmed the vector boundary of Nanning in ArcGIS to obtain the LU data of Nanning in the third phase (Figure 2).

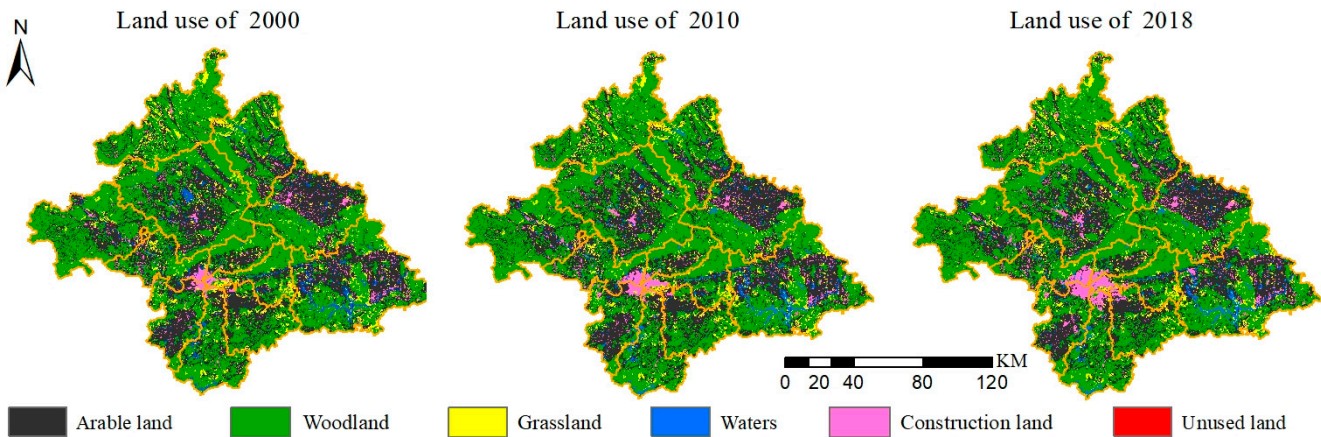

**Figure 2.** LU of Nanning in different years.

Meteorological data were obtained from the European center for medium-range weather forecasts (ECMWF) reanalysis data of the third generation ERA—Interim daily/levtype = SFC (https://apps.ecmwf.int/ (accessed on 16 December 2020)), with a spatial resolution of 0.5°. ERA-Interim uses mind-variational analysis, improved humidity analysis, and error correction of satellite data to improve the quality of the reanalysis data. The GLASS FVC dataset was downloaded from Beijing Normal University (http://glass-product.bnu.edu.cn/ (accessed on 27 December 2021)). FVC data were obtained by training a generalized regression neural network model using high-precision Landsat TM/ETM+ data and MODIS data. The spatial resolution is 500 m, and the temporal resolution is 8d. The digital elevation model (DEM) data were obtained from China Geospatial Data Cloud (http://www.gscloud.cn/ (accessed on 7 March 2021)), with a spatial resolution of 30 m. The DEM data were spliced, clipped, and reprojected, and the elevation and slope were selected to discuss their influence on LU change.

2.2.2. Other Data

Data on leaf area index, gross domestic product (GDP), and population density were obtained from the Resources and Environmental Sciences and Data Center, Chinese Academy of Sciences, with the spatial resolution of 8 km, 1 km, and 1 km, respectively. The road data were obtained from the China Geographic Information Resources Directory Service System (https://www.webmap.cn/ (accessed on 14 January 2021)), and the railway and expressway data were selected according to the research needs.

*2.3. Research Methodology*

This study explores the scale implications of ecological risk from the spatial granularity and extent perspectives using three phases of LU data from 2000, 2010, and 2018 in Nanning, China, and determines the optimal scale for an ERA in Nanning. To this end, the LU transfer matrix and ERA model were used to analyze the characteristics of spatial and temporal changes in LU and ecological risk in Nanning under the optimal scale. Additionally, the PLUS model was used to simulate the ecological risk in Nanning during the year 2036 under two scenarios of natural development and ecological preservation given ten socioeconomic and human land-use factors. The results were combined to provide a reference for ecological risk avoidance and ecological restoration in Nanning. Figure 3 provides a flowchart illustrating the particular research methods and content of the study.

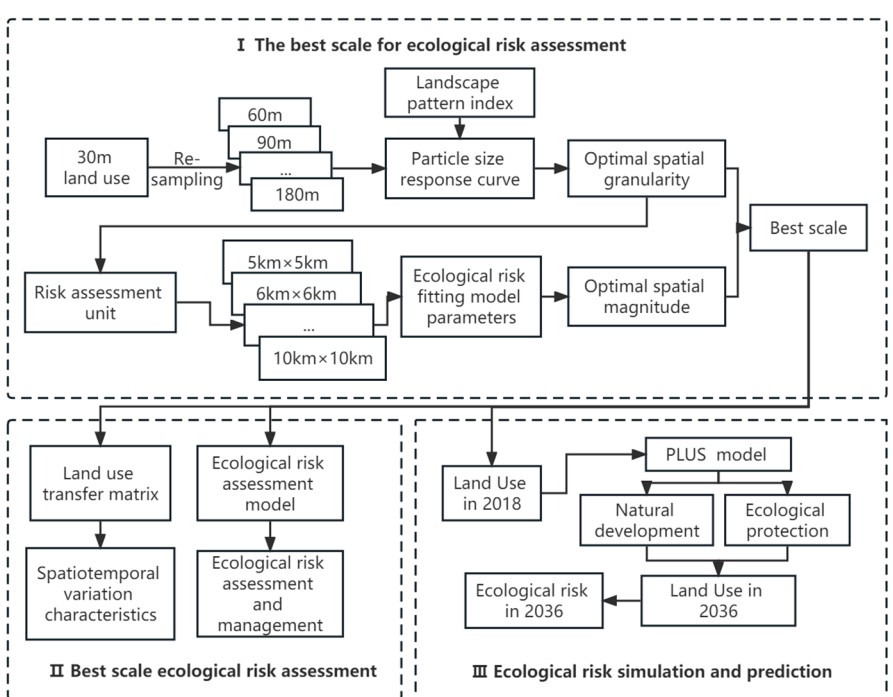

**Figure 3.** The framework of this research.

2.3.1. Scale Conversion

In landscape ecology, the spatial scale is typically divided into spatial granularity and extent [43]. Spatial granularity refers to a feature's length, area, or volume and is represented by the smallest identifiable unit in the landscape. Based on the findings of previous studies [44], this study set the granularity range between 30 and 180 m, resampled 30 m LU data at 30 m intervals using ArcGIS, calculated each landscape pattern index using Fragstats 4.1, and plotted the response of each landscape pattern index under six different granularity levels from 2000 to 2018. The optimal landscape grain size in Nanning was determined using the inflection points and relaxation intervals in the response curve.

Spatial extent refers to the area of the study region. Previous studies have demonstrated that when the grid size is 2 to 5 times the average patch area, the integrity of patches

and spatial differentiation in the ecological risks can be maintained [45]. Therefore, in this study, ArcGIS was used to create a fishing grid tool to generate seven types of square grid ERA cells of different sizes: 4 km × 4 km, 5 km × 5 km, 6 km × 6 km, 7 km × 7 km, 8 km × 8 km, 9 km × 9 km, and 10 km × 10 km. The ecological risk index value of each cell grid was determined using Fragstats4.1 software, and the ERA model is regarded as the ecological risk value of each grid cell center. The results fitting the ecological risk semi-variance function under different extents in Nanning from 2000 to 2018 were analyzed to determine the best scale for the ERA and prediction.

### 2.3.2. Landscape Pattern Index Selection

In order to comprehensively reflect the characteristics of landscape patterns, scholars typically select a number of landscape pattern indices from the following four aspects: area, form, distribution, and diversity. Based on previous studies [46], seventeen landscape pattern indices were selected in this study. Among them, the area edge indices include the total area (TA), largest patch index (LPI), edge density (ED), and mean patch area (AREA_MN). The shape indices include the mean shape index (SHAPE_MN), mean fractal dimension index (FRAC_MN). The dispersion indices include the number of patches (NP), patch density (PD), landscape division index (DIVISION), splitting index (SPLIT), contagion (CONTAG), interspersion juxtaposition index (IJI), aggregation index (AI), landscape shape index (LSI), and patch cohesion index (COHESION). The diversity indices include the Shannon's diversity index (SHDI) and Shannon's evenness index (SHEI).

The coefficient of variation $C_v$ is used to measure the sensitivity of each landscape pattern index to changes in grain size, and the appropriate landscape pattern index can be selected for the analysis of optimum grain size according to the sensitivity degree [47]. The calculation formula is as follows:

$$C_V = \frac{1}{\overline{x}} \times \sqrt{\frac{1}{n}\sum_{j=1}^{n}(x_j - \overline{x})^2} \times 100\% \tag{1}$$

In the formula, $C_v$ is the coefficient of variation; $x_j$ is the landscape index value at $j$ granularity; $\overline{x}$ is the average value of the landscape index at different granularities; and $n$ is the number of granularity grades. According to its size, the variation coefficient is divided into five grades: insensitive, low sensitivity, medium sensitivity, high sensitivity, and extremely high sensitivity. The assignment interval is insensitive (<1%), low sensitivity (1~4%), medium sensitive (4~7%), highly sensitive (7~10%), and extremely sensitive (>10%). To avoid duplication and invalidity in the selection of landscape pattern indices, the ones insensitive to spatial grain size changes were not involved in the grain size change analysis.

### 2.3.3. Semi-Variance Functions

The semi-variance function in geostatistics is a function that exposes the variance value of data points and the separation between them, and it is mostly used to explain and detect the spatial structure of patterns as well as for spatially local optimum interpolation [48]. To determine the optimal extent and model for kriging interpolation, based on the results fitting the semi-variance function to different gridded ERA cells, the calculation formula is as follows:

$$\gamma(h) = \frac{1}{2N(h)} \sum_{i=1}^{N(h)} [Z(x_i) - Z(x_i + h)]^2 \tag{2}$$

In the formula, $\gamma(h)$ is the semi-variance function; $h$ is the spatial distance between samples; $Z(x_i)$ and $Z(x_i + h)$ are the values of the variables at positions xi and $x_i + h$, respectively, $i = 1, 2, \ldots, N(h)$; and $N(h)$ is the total number of samples when the spatial distance is $h$. When $h = 0$ and $\gamma(h) = C0$, C0 is called the block gold value, indicating the ecological risk is the spatial heterogeneity generated by random factors; C0 + C is the abutment value, indicating the overall spatial heterogeneity of the autocorrelated

part of the ecological risk; A0 is called the variation range, that is, the sampling distance corresponding to the first time the semi-variance function reaches a stable value, indicating the autocorrelated range of the ecological risk.

GS + 9.0 software was used to calculate the block gold values, variance, residuals, and complex correlation coefficients of the semi-variance functions of ecological risks in Nanning for three periods under different extents in 2000, 2010, and 2018, and to analyze the spatial heterogeneity in ecological risks under different extents. Lastly, we use the results to decide which scale will be most effective in Nanning for evaluating ecological risk.

### 2.3.4. LU Change

The LU transfer matrix represents the quantitative relationship between the LU-type conversion in two different periods in the form of a matrix [49]. The calculation formula is as follows:

$$S_{ij} = \begin{bmatrix} S_{11} & S_{12} & \dots & S_{1n} \\ S_{21} & S_{22} & \dots & S_{2n} \\ \dots & \dots & \dots & \dots \\ S_{n1} & S_{n2} & \dots & S_{nn} \end{bmatrix} \tag{3}$$

where $S_{ij}$ represents the amount from the initial type $i$ to the final type $j$, and $i$ and $j$ represent the LU types at the initial and final stages of the study, respectively. $n$ represents the total number of LU types.

### 2.3.5. ERA Models

To establish the link between ecological risk and landscape structure, the landscape disturbance index ($E_i$) and the landscape vulnerability index ($F_i$) were selected in this study to construct a comprehensive Ecological Risk Index (*ERI*) model for Nanning. The ERI was calculated using the following formula:

$$ERI_k = \sum_{i}^{N} \frac{S_{ki}}{S_k} \sqrt{E_i \times F_i} \tag{4}$$

where $ERI_k$ represents the *ERI* of the $k$-th risk plot; $S_{ki}$ represents the area of the $k$-th risk plot of the $i$-th landscape type; $S_k$ denotes the total area of the $k$-th risk plot; $E_i$ indicates the landscape disturbance index for the $i$-th landscape type; and $F_i$ denotes the landscape vulnerability index for the $i$-th landscape type. The calculation formulas and ecological significance of $E_i$ and $F_i$ are derived from previous studies [49].

According to the ecological risk calculation results, it can be seen that the ecological risk values of Nanning from 2000 to 2018 ranged from 9.19 to 13.09. To further analyze the spatial and temporal variation characteristics of ecological risk in Nanning, the natural breakpoint method was implemented in ArcGIS to classify the landscape ecological risk of the study area into five levels: low-risk (<10.23), medium-low-risk (10.23~10.75), medium-risk (10.75~11.27), medium-high-risk (11.27~11.88), and high-risk (>11.88). Meanwhile, areas with reducing ecological risk levels were set as improvement areas, areas with unchanged ecological risk levels as stable areas, and areas with increasing ecological risk levels as deterioration areas.

### 2.3.6. The PLUS Model

The PLUS model is a meta-cellular automata CA model that combines the land expansion analysis strategy (LEAS) and the multi-class random patch seed (CARS) model, reflecting the driving factors of LU change and their contribution rates with good LU simulation accuracy [38].

(1)     Selection of driving factors for LU

LU change results from a combination of natural and socio-economic factors. Based on the current ecological situation and data acquisition in Nanning, ten factors were selected in both natural and socio-economic aspects: slope, elevation, precipitation, temperature, FVC, population density, leaf area index, GDP, distance from highways, and distance from railways.

(2)     Neighborhood parameter settings

The neighborhood weight parameter is an important indicator reflecting the expansion intensity of different LU types. The parameter ranges from 0 to 1, and the closer it is to 1, the greater the expansion intensity of the LU type. In this study, the neighborhood parameters of arable land, woodland, grassland, waters, construction land, and unused land are assigned values of 0.7, 0.4, 0.3, 0.2, 0.2, 0.9, and 0.2.

(3)     Scenario setting

Regional LU changes are different under various scenarios, and simulating the future LU of Nanning under different scenarios provides different reference perspectives for decision-makers to plan the national land space. In this study, according to the LU change characteristics of Nanning and previous studies [50], two scenarios, namely the natural development scenario (NDS) and the ecological protection scenario (EPS), were set to simulate and forecast the LU of Nanning in 2036. Among them, the NDS refers to the transfer cost matrix based on the LU transfer matrix and transfer probability of Nanning from 2000 to 2018, and the EPS refers to the restriction of the transfer of high-quality LU. The transfer cost matrix corresponding to each scenario is as follows (Table 1), where 0 means no transfer is allowed and 1 means transfer is allowed.

(4)     Model accuracy verification

To ensure the accuracy of the simulation results, the Kappa coefficient and overall accuracy were used to test the simulation results. In this study, the 2018 LU data were obtained with simulation using the 2000 LU data. By comparing the 2018 LU simulation data with the actual data, it was found that the Kappa coefficient was 0.82, and the overall accuracy was 85.41%.

**Table 1.** Simulation cost matrix of each scenario.

| 2018–2036 | NDS | | | | | | EPS | | | | | |
|---|---|---|---|---|---|---|---|---|---|---|---|---|
| | T1 | T2 | T3 | T4 | T5 | T6 | T1 | T2 | T3 | T4 | T5 | T6 |
| T1 | 1 | 1 | 1 | 1 | 1 | 0 | 1 | 1 | 1 | 0 | 1 | 0 |
| T2 | 1 | 1 | 1 | 0 | 1 | 0 | 0 | 1 | 1 | 0 | 0 | 0 |
| T3 | 1 | 1 | 1 | 1 | 1 | 0 | 0 | 1 | 1 | 0 | 0 | 0 |
| T4 | 1 | 1 | 1 | 1 | 1 | 1 | 0 | 0 | 1 | 1 | 0 | 0 |
| T5 | 1 | 1 | 0 | 0 | 1 | 0 | 1 | 0 | 1 | 0 | 1 | 0 |
| T6 | 0 | 0 | 0 | 1 | 0 | 1 | 1 | 1 | 1 | 1 | 1 | 1 |

Note: T1, T2, T3, T4, T5, and T6 represent arable land, woodland, grassland, waters, construction land, and unused land, respectively.

## 3. Results and Analysis

### 3.1. Optimal Scale Analysis of Ecological Risk

#### 3.1.1. Optimal Granularity Analysis of Landscape Pattern Index

The sensitivity of each landscape pattern index to spatial granularity change remained broadly consistent from 2000 to 2018 (Figure 4). The SPLIT index was susceptible to spatial granularity change, with the SHAPE_MN and CONTAG indices as high-sensitivity indicators, the LPI, ED, LSI, and AI indices as medium-sensitivity indicators, and the four landscape pattern indices NP, PD, AREA_MN, and FRAC_MN have low sensitivity to

spatial granularity changes. The coefficient of variation values of TA, IJI, COHESION, DIVISION, SHDI, and SHEI indices are all less than 1%.

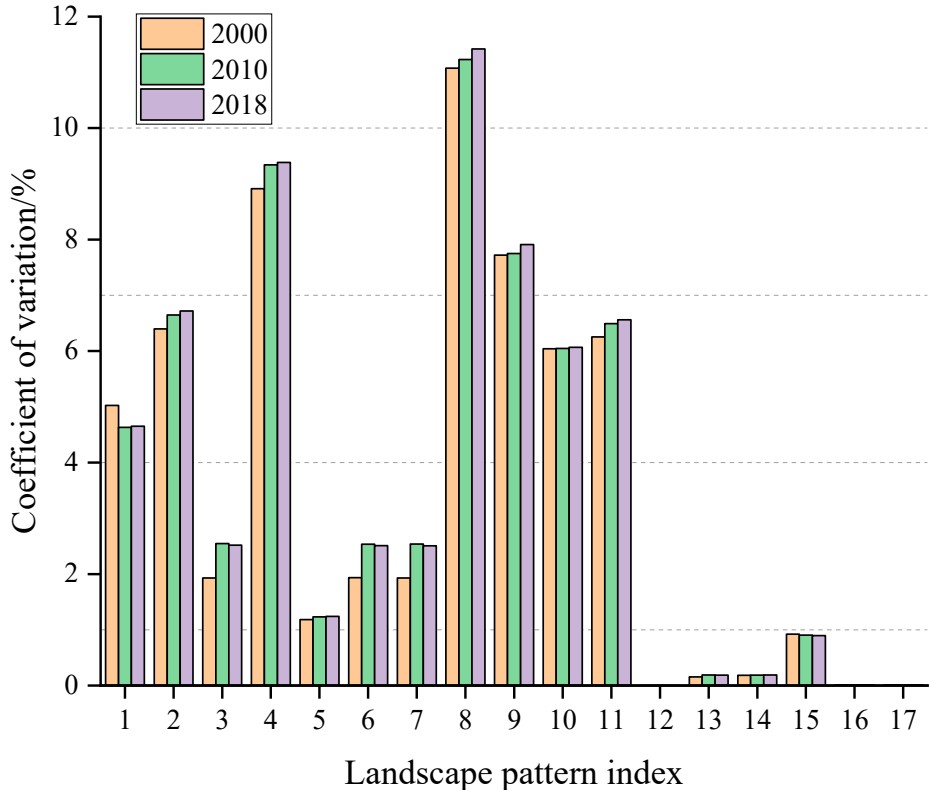

**Figure 4.** Variation coefficient of the landscape pattern index in response to spatial granularity. Note: 1–17 represent LPI, ED, AREA_MN, SHAPE_MN, FRAC_MN, NP, PD, SPLIT, CONTAG, AI, LSI, TA, IJI, COHESION, DIVISION, SHDI, and SHEI, respectively.

The response of different landscape pattern indices to the spatial granularity change was different in the same period (Figure 5). Among them, the grain size response curves of ED, SHAPE_MN, FRAC_MN, CONTAG, AI, and LSI showed no obvious inflection point and gradually decreased with the increase of spatial granularity. The responses of the NP and PD indices to the changes in spatial granularity were the same, with a slight increase in the 30–60 m particle size interval, a sharp increase in the 60–120 m particle size interval, a gentle change in the 120–150 m particle size interval without noticeable fluctuation, and a downward trend in the 150–180 m particle size interval. The AREA_MN index has no apparent fluctuation in the 120–150 m particle size interval, but the response to the spatial particle size change in other intervals is completely opposite to the NP index and PD index trends. Therefore, 120 m was selected as the best landscape granularity in Nanning using a comprehensive analysis of the landscape pattern index's inflection points and relaxation intervals.

### 3.1.2. Analysis of the Optimal Extent of Ecological Risk

The fitting model parameters of the ecological risk semi-variogram in different years from 2000 to 2018 showed the same trend with the increase in extent, but the response of the fitting model parameters of the semi-variogram in the same year to the spatial extent was different (Table 2). When the extent is 4–7 km, $C_0$ shows a decreasing trend. When the extent is 7–10 km, $C_0$ shows fluctuations described as rising first, then falling, then rising again, and $C_0$ reaches a minimum value at 9 km. $C_0 + C$ decreases gradually when the extent is 4–7 km. When the extent is 7–10 km, $C_0 + C$ increases slightly without noticeable fluctuation. When the extent is changed from 4 to 7 km, $C/(C_0 + C)$ shows a gradually increasing trend, and when it goes from 7 to 10 km, $C/(C_0 + C)$ shows a decreasing trend

first, followed by an increasing and then a decreasing trend again. C/(C0 + C) reaches a maximum value at 9 km. When the spatial extent is 7 km, RSS is the smallest, and $R^2$ is the largest. Therefore, 7 km is selected as the best scale for ERA and prediction.

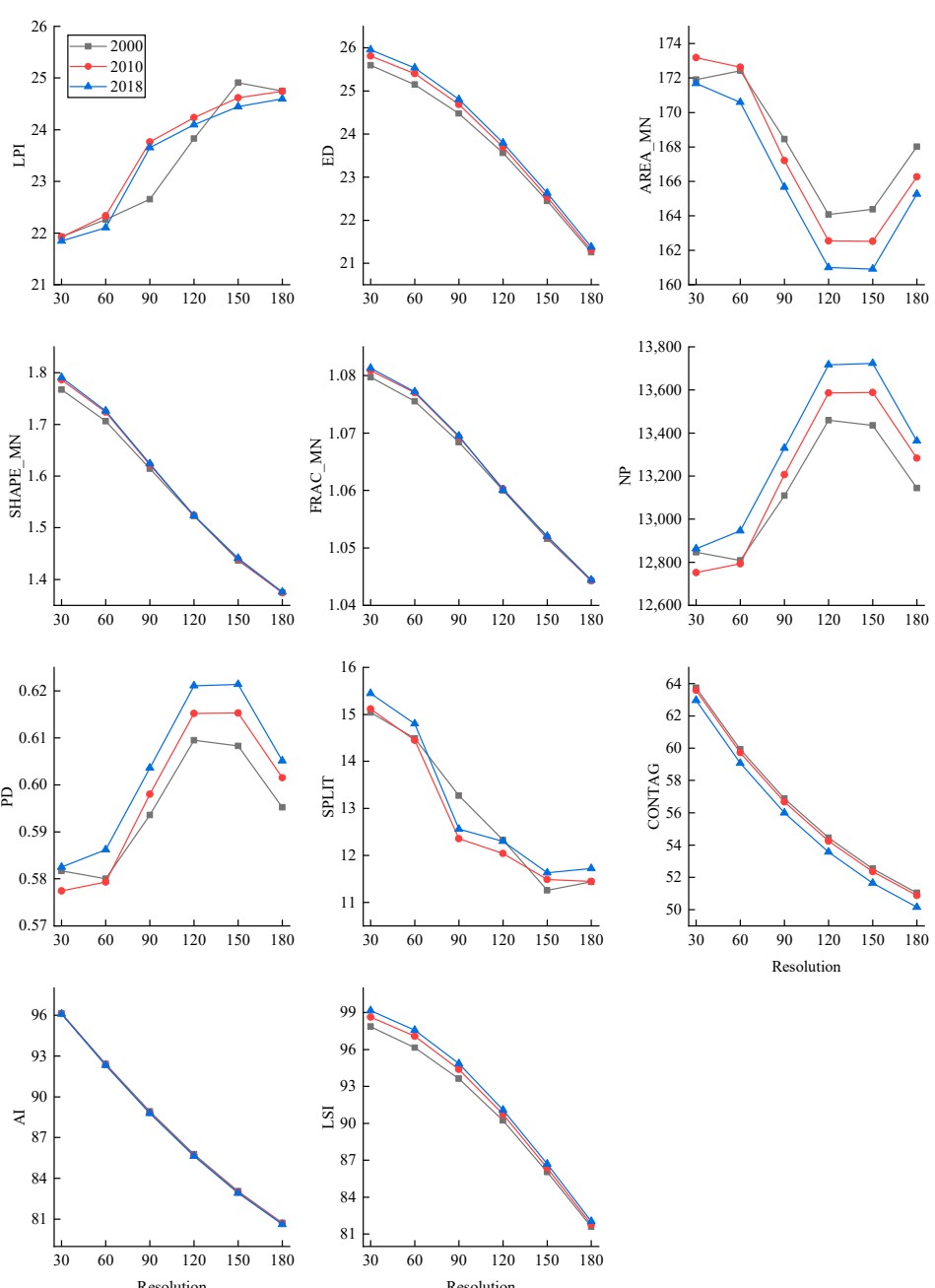

**Figure 5.** Granularity effect of the landscape pattern index.

**Table 2.** Fitting model parameters of the ecological risk semi-variogram under different extents.

| Spatial Extent/km | Year | C0 | C0 + C | C/(C0 + C) | A/m | RSS | $R^2$ |
|---|---|---|---|---|---|---|---|
| 4 | 2000 | 0.652 | 2.098 | 0.689 | 31,900 | 0.0250 | 0.975 |
| | 2010 | 0.709 | 2.084 | 0.660 | 32,900 | 0.0557 | 0.944 |
| | 2018 | 0.690 | 1.974 | 0.650 | 31,400 | 0.0466 | 0.943 |
| 5 | 2000 | 0.399 | 1.881 | 0.788 | 32,300 | 0.0431 | 0.965 |
| | 2010 | 0.437 | 1.85 | 0.764 | 32,600 | 0.106 | 0.919 |
| | 2018 | 0.431 | 1.767 | 0.756 | 31,400 | 0.0921 | 0.918 |

**Table 2.** *Cont.*

| Spatial Extent/km | Year | C0 | C0 + C | C/(C0 + C) | A/m | RSS | R$^2$ |
|---|---|---|---|---|---|---|---|
| 6 | 2000 | 0.3 | 1.600 | 0.813 | 32,100 | 0.0189 | 0.976 |
| | 2010 | 0.204 | 1.631 | 0.875 | 27,800 | 0.0474 | 0.958 |
| | 2018 | 0.294 | 1.455 | 0.798 | 31,100 | 0.0245 | 0.96 |
| 7 | 2000 | 0.208 | 1.447 | 0.856 | 33,500 | 0.0560 | 0.976 |
| | 2010 | 0.165 | 1.352 | 0.878 | 32,700 | 0.0331 | 0.959 |
| | 2018 | 0.172 | 1.277 | 0.865 | 31,800 | 0.0199 | 0.959 |
| 8 | 2000 | 0.291 | 1.481 | 0.804 | 39,000 | 0.0245 | 0.969 |
| | 2010 | 0.349 | 1.477 | 0.764 | 40,900 | 0.0682 | 0.927 |
| | 2018 | 0.331 | 1.382 | 0.76 | 39,300 | 0.065 | 0.916 |
| 9 | 2000 | 0.123 | 1.477 | 0.917 | 36,400 | 0.0333 | 0.962 |
| | 2010 | 0.177 | 1.487 | 0.881 | 36,700 | 0.0538 | 0.908 |
| | 2018 | 0.192 | 1.379 | 0.861 | 36,200 | 0.0429 | 0.938 |
| 10 | 2000 | 0.218 | 1.504 | 0.855 | 35,000 | 0.0667 | 0.85 |
| | 2010 | 0.323 | 1.517 | 0.787 | 36,600 | 0.1170 | 0.753 |
| | 2018 | 0.290 | 1.400 | 0.793 | 35,200 | 0.0972 | 0.745 |

*3.2. Analysis of Temporal and Spatial Variation Characteristics of LU and Ecological Risk at the Optimal Scale*

3.2.1. Analysis of Temporal and Spatial Changes in LU

The area size of each LU type in Nanning was woodland > arable land > grassland > construction land > waters > unused land (Table 3). The woodland and arable land area accounted for 52.80% and 34.00% of the total area, respectively, and the construction land area was about 973.31 km$^2$. From 2000 to 2018, the area of arable land and unused land decreased first and then increased, while the area of woodland, grassland, and water continued to decrease, and the area of construction land increased significantly. Among them, the area of construction land and unused land increased by 336.33 km$^2$ and 1.44 km$^2$, respectively, while the area of arable land, woodland, grassland, and waters decreased by 206.32 km$^2$, 51.39 km$^2$, 48.67 km$^2$, and 30.79 km$^2$, respectively.

**Table 3.** Areas of LU types in Nanning from 2000 to 2018 (km$^2$).

| Land-Use Types | 2000 | 2010 | 2018 |
|---|---|---|---|
| Arable land | 7592.08 | 7547.30 | 7385.76 |
| Woodland | 11,663.80 | 11,663.63 | 11,612.40 |
| Grassland | 1350.19 | 1322.44 | 1301.52 |
| Waters | 648.20 | 621.06 | 617.41 |
| Construction Land | 827.60 | 928.40 | 1163.92 |
| Unused land | 2.29 | 1.93 | 3.73 |

From 2000 to 2018, the area converted by different LU types reached 630.22 km$^2$ (Table 4). The area of arable land transferred out was the largest, reaching 318.48 km$^2$ and accounting for 50.54% of the total area transferred, mainly converted to construction land and woodland. Secondly, the transferred area of woodland was more extensive reaching 155.07 km$^2$. The transfer area of construction land was the largest, reaching 359.04 km$^2$, which was mainly converted from arable land and woodland.

**Table 4.** Transfer matrix of LU types in Nanning from 2000 to 2018 (km$^2$).

| 2000 | 2018 | | | | | | |
|---|---|---|---|---|---|---|---|
| | **Arable Land** | **Woodland** | **Grassland** | **Waters** | **Construction Land** | **Unused Land** | **Sum** |
| Arable land | 7273.60 | 59.27 | 9.09 | 10.12 | 239.85 | 0.16 | 7592.08 |
| Woodland | 54.66 | 11,508.72 | 21.12 | 7.75 | 71.52 | 0.01 | 11,663.80 |
| Grassland | 7.57 | 32.52 | 1268.57 | 1.56 | 39.97 | 0 | 1350.19 |
| Waters | 33.49 | 7.00 | 2.10 | 596.26 | 7.69 | 1.66 | 648.20 |
| Construction Land | 16.21 | 4.54 | 0.62 | 1.34 | 804.89 | 0 | 827.60 |
| Unused land | 0 | 0 | 0 | 0.39 | 0 | 1.911 | 2.29 |
| Sum | 7385.54 | 11,612.04 | 1301.50 | 617.41 | 1163.92 | 3.73 | 22,084.16 |

### 3.2.2. Ecological Risk Change

From 2000 to 2018, the ecological risks in Nanning were mainly low-medium, medium-high, and medium, accounting for 24.35%, 22.77%, and 22.04% of the total area of ecological risks in Nanning, respectively. The low-risk and high-risk areas were relatively small (Figure 6). From 2000 to 2010, the area of each grade of ecological risk changed slightly, but in 2018, each grade of ecological risk changed significantly. The area variation trend of different ecological risk levels was different. The low-risk, low-medium-risk, and medium-risk areas all showed an increasing trend, while the medium-high-risk and high-risk areas decreased. Among them, the medium-risk area increased the most, reaching 683.856 km$^2$, while the area of high-risk decreased the most, reaching 839.95 km$^2$.

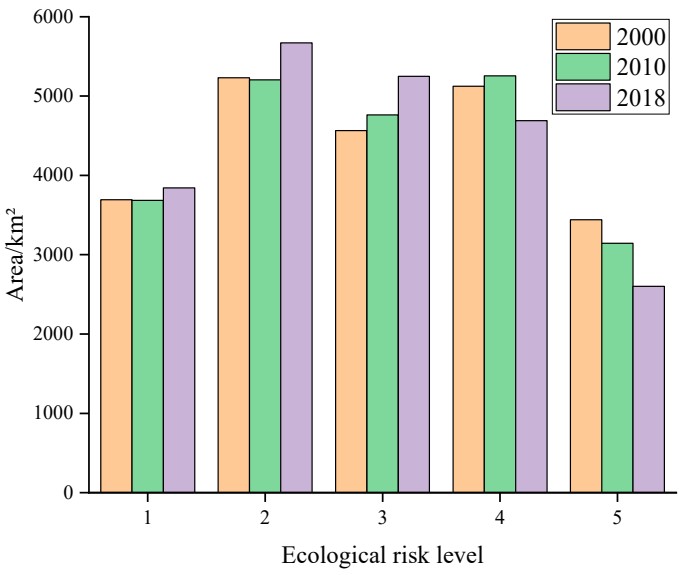

**Figure 6.** Area map of the ecological risk level in Nanning. Note: 1, 2, 3, 4, and 5 represent low-risk, low-medium-risk, medium-risk, medium-high-risk, and high-risk, respectively.

From 2000 to 2018, Nanning's overall ecological risk level was not high, but there were significant differences in the spatial distribution of each level of ecological risk (Figure 7). The spatial distribution of each ecological risk level was centered in the high-risk areas and expanded to the low-risk ones according to the ecological risk level. The high-risk areas are mainly distributed in central Wuming, northern Binyang, central Hengzhou, and northern Yongning. The low-risk areas are mainly distributed in Mashan, the west of Longan, and the junction of Xingning, Qingxiu, and Binyang counties in the middle of Nanning. The high-risk and medium-high-risk areas showed a significant decreasing trend, mainly distributed in the Xixiangtang, Jiangnan, Liangqing, and Yongning areas in the southwest of Nanning.

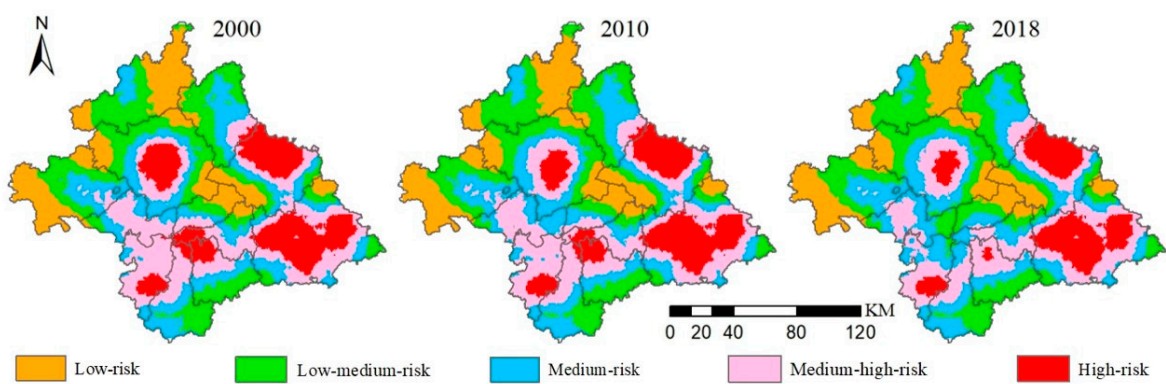

**Figure 7.** Spatial distribution of ecological risk levels in Nanning.

### 3.2.3. Distribution of Ecological Risk Land Types

The area distribution of different LU types in each level of ecological risk was different in Nanning (Figure 8). The arable land was mainly distributed in the medium-high-risk and high-risk areas. Compared with 2010, in 2018, the arable land area in the medium-high-risk and high-risk areas decreased by 65.84 km$^2$ and 478.25 km$^2$, respectively, while the arable land area in other ecological risk levels showed an increasing trend. The overall ecological risk in woodland and grassland is low, and the proportion of woodland and grassland areas distributed in high-risk areas is small. In 2000, the water area was mainly distributed in the high-risk area, while in 2018, it was mainly distributed in the medium-high-risk area. In 2018, the main distribution area of construction land decreased from medium-high-risk in 2000 to medium-risk. Compared with 2010, in 2018, the unused land area in the high-risk areas increased by 0.66 km$^2$. In general, the high-risk water region area increased while the area of other LU types in the high-risk regions decreased continuously.

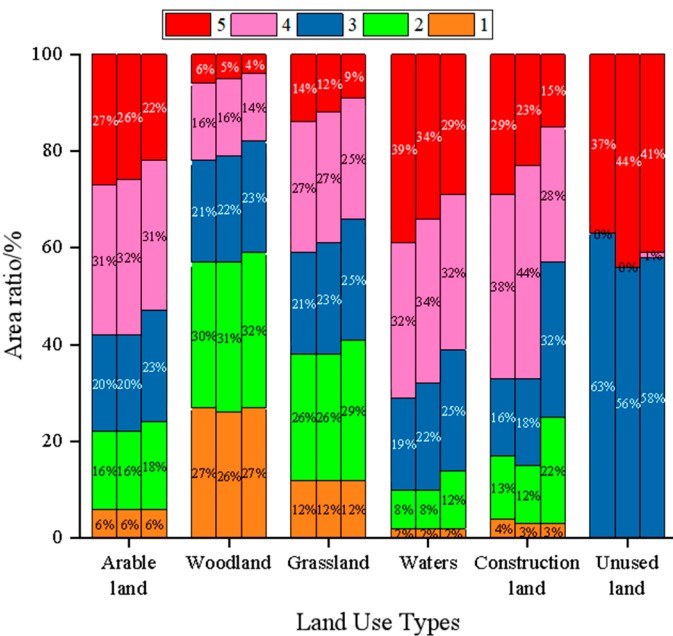

**Figure 8.** Distribution of ecological risks by land-use types. Note: 1, 2, 3, 4, and 5 represent low-risk, low-medium-risk, medium-risk, medium-high-risk, and high-risk, respectively. The corresponding year of ecological risk for each land use type is 2000, 2010, and 2018 from left to right.

### 3.3. Ecological Risk Simulation and Prediction

### 3.3.1. Ecological Risk Prediction in Different Scenarios

Compared with 2018, the ecological risks under the two scenarios in Nanning in the year 2036 were still dominated by low-medium and medium risks, but the areas of

ecological risks at each level were different (Table 5). Compared with 2018, the medium-high-risk region areas decreased by 486.13 km$^2$ in 2036 under the NDS, while the other region areas increased, and the low-risk and high-risk region areas increased significantly by 293.57 km$^2$ and 114.58 km$^2$, respectively. In 2036, the ecological risk of Nanning was improved under the EPS. The medium-high and the high-risk region areas showed a decreasing trend, reducing to 430.47 km$^2$, while the low, low-medium, and medium risk region areas increased. In 2036, the ecological risk of Nanning under the EPS was significantly lower than that under the NDS, and the area of the medium-high risk and high-risk regions decreased by 58.92 km$^2$.

**Table 5.** Comparison of ecological risk level area under different scenarios (km$^2$).

| Year | Scenario | Low-Risk | Low-Medium-Risk | Medium-Risk | Medium-High-Risk | High Risk |
|---|---|---|---|---|---|---|
| 2018 | | 3841.10 | 5670.81 | 5249.15 | 4689.27 | 2600.53 |
| 2036 | NDS | 4134.67 | 5680.27 | 5316.57 | 4203.14 | 2715.11 |
| | EPS | 4049.74 | 5786.77 | 5353.92 | 4297.75 | 2561.57 |
| 2018–2036 | NDS | 293.57 | 9.46 | 67.42 | −486.13 | 114.58 |
| | EPS | 208.64 | 115.96 | 104.77 | −391.52 | −38.95 |

The spatial distribution pattern of overall ecological risks in 2036 was roughly similar to that in 2018 under the two scenarios (Figure 9). Compared with the ecological risk in 2018, under the NDS, the high-risk region area in central Wuming and north Yongning decreased in 2036, while the high-risk region area in Hengzhou showed an increasing trend. Under the EPS, the high-risk region areas in central Wuming, north Yongning, and north Jiangnan decreased. When comparing the ecological hazards under the two scenarios for Nanning in 2036, the ecological risk under the NDS peaked at 13.11, while the ecological risk under the EPS peaked at 13.05. Under the NDS, the area of the high-risk regions was much larger than it was under the EPS.

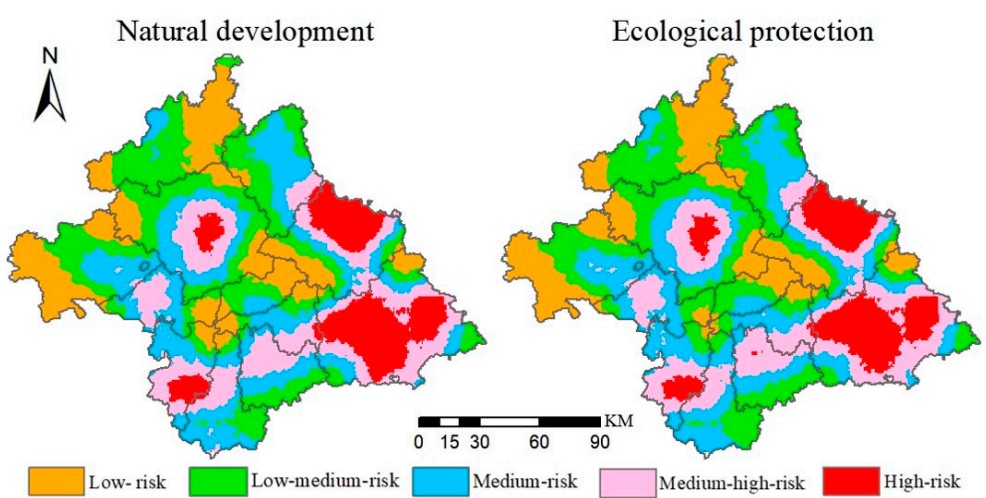

**Figure 9.** The ecological risk prediction for Nanning in 2036.

3.3.2. Spatial Changes in Ecological Risks under Different Scenarios

From 2018 to 2036, the changes in the ecological risk level in the two scenarios in Nanning were basically the same in general location, but the area changes differed (Figure 10). Under the two scenarios, the ecological risk level improvement areas were mainly distributed at the junction of Xixiangtang, Jiangnan, Liangqing, Yongning, Qingxiu, and Xingning districts. In contrast, the ecological risk level deterioration areas were scattered to the north and south of Nanning. Among them, the area of ecological risk grade improvement under

the NDS and EPS were 1832.85 km$^2$ and 1524.15 km$^2$, respectively. In comparison, the area of ecological risk grade deterioration under the NDS and EPS were 1253.36 km$^2$ and 533.7936 km$^2$, respectively.

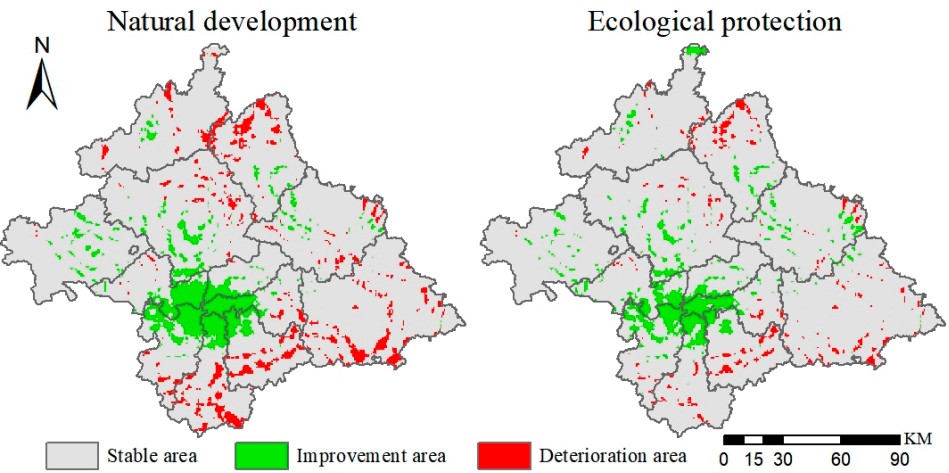

**Figure 10.** Changes in ecological risk in Nanning under different scenarios from 2018 to 2036.

## 4. Discussion

### 4.1. Analysis of the Impact of Scale Effect on Ecological Risk

Scale has always been the focus and difficulty of researching geography, ecology, and related disciplines. Scale effects run through many research fields, such as landscape patterns and processes, ecosystem structure and function, and topographic characteristics [51]. Different disciplines have different understandings of the scale concept and content. Landscape ecological risk research divides scale into spatial granularity and extent, considering its typical spatial heterogeneity and scale dependence. Current research focuses on the granularity of the landscape pattern index response and less on the two aspects of spatial granularity and extent to determine the best measure of landscape ecological risk. Thus, this research uses the granularity of the response curve to determine the best landscape pattern index granularity, based on the fitting results of ecological risk area semi-variance functions, to choose the best spatial extent and to determine the best measure of ERA in Nanning.

The sensitivity of different landscape pattern indices to changes in spatial granularity in different regions varies. According to the coefficient of variation of landscape pattern indices to grain size changes in Nanning, the SPLIT, SHAPE_MN, and CONTAG indices have high sensitivity to the changes in spatial grain size, which is due to the changes in data raster cells caused by the changes in spatial scale. The shape and aggregation degree in the plaque changes significantly. With the spatial granularity change, the internal structure of the landscape and its pattern index change accordingly [52]. The increase in spatial granularity changes the complexity of landscape patch boundaries, leading to the aggregation of some patches and the gradual merging of small patches with other patches, resulting in the reduction in the number and aggregation of patch boundaries and the simplification of patch shape [53]. Different landscape pattern indices have different responses to spatial grain size changes, so it is very difficult to find an optimal spatial grain size conforming to all indexes. In practical application, we balance the sensitivity of each index to spatial grain size to determine the optimal spatial grain size. Based on the turning point and gentle interval of each landscape pattern index grain size response curve, 120 m is identified as the inflection point of most landscape pattern response curves and is in the moderate interval. Therefore, this study believes 120 m is the best landscape spatial granularity in Nanning.

The appropriate spatial extent can directly reflect the changed law of regional landscape patterns and improve the accuracy and scientific validity of ERA. Comprehensively

considering the turning and extreme points of the parameters in the semi-variogram fitting model of ecological risks in Nanning city under different amplitudes, it can be seen that when the space range is 7 km, C0, and C0 + C is a significant turning point and 7 km is the minimum. This shows that 7 km is a random part of the minimum point of spatial heterogeneity, and the ecological risk of variations in the amplitude are stable. At the same time, the fitting accuracy of the ecological risk semi-variogram in Nanning was highest when the spatial amplitude was 7 km, so this study believed that 7 km was the best spatial extent in Nanning.

### 4.2. Analysis of Temporal and Spatial Changes in Ecological Risk

As a typical landscape garden city in China, Nanning is rich in mountains, forests, fields, lakes, and grass resources, but the rapid development of urban construction has led to many ecological contradictions and problems. There are pronounced regional differences in the results of the ERA in Nanning, which is due to the obvious differences in natural conditions and human activities in different regions. Different natural environments determine human activities, and human activities, in turn, change the regional natural environment. The woodland area of Nanning is wide, concentrated in the northern and marginal areas of the city, where the temperature is suitable, the precipitation is abundant, the vegetation coverage is high, and the ecological risk is relatively low. The seven districts under the jurisdiction of Nanning have high temperatures, low precipitation, and flat terrain. The area has a large population, a high GDP, and frequent human activities, and the construction land is distributed chiefly here, so the ecological risk is relatively high. Water resources in Hangzhou and Binyang are widely distributed, but the water quality is poor, the pollution problem is serious, and the scattered distribution of construction land in this area leads to a higher ecological risk.

The change in LU and human activities influences spatial and temporal changes in ecological risk. From 2000 to 2018, the ERA of Nanning was significantly improved, and the area of high-risk and medium-high-risk areas decreased significantly, totaling 1272.72 km$^2$, mainly distributed in the seven districts under the jurisdiction of Nanning, which was related to the change of construction land in this region. From 2000 to 2018, the construction land expanded significantly, reaching 336.33 km$^2$, and its fragmentation and separation indexes decreased significantly (Table 6), indicating that the continuous and contiguous development of construction land enhanced the degree of aggregation, increased internal stability, and gradually reduced ecological risk. In recent years, Nanning has carried out a series of ecological construction and ecological restoration work, such as remediation of black and smelly water, soil pollution prevention and control, and rural environment treatment, which have achieved remarkable results and improved the ecological risk of Nanning to a certain extent.

**Table 6.** Landscape pattern indices for different land-use types.

| Type | Year | NP | $C_i$ | $DO_i$ | $S_i$ | $E_i$ | $F_i$ |
|---|---|---|---|---|---|---|---|
| Arable land | 2000 | 5584 | 0.0074 | 0.2945 | 0.0731 | 0.0852 | 0.1905 |
| | 2010 | 5629 | 0.0075 | 0.2971 | 0.0739 | 0.0853 | 0.1905 |
| | 2018 | 5764 | 0.0078 | 0.2946 | 0.0764 | 0.0857 | 0.1905 |
| Woodland | 2000 | 3436 | 0.0029 | 0.3647 | 0.0373 | 0.0856 | 0.0952 |
| | 2010 | 3424 | 0.0029 | 0.3642 | 0.0373 | 0.0855 | 0.0952 |
| | 2018 | 3400 | 0.0029 | 0.3623 | 0.0373 | 0.0851 | 0.0952 |
| Grassland | 2000 | 3161 | 0.0234 | 0.1216 | 0.3094 | 0.1288 | 0.1429 |
| | 2010 | 3212 | 0.0243 | 0.1212 | 0.3184 | 0.1319 | 0.1429 |
| | 2018 | 3216 | 0.0247 | 0.1203 | 0.3238 | 0.1335 | 0.1429 |
| Waters | 2000 | 2372 | 0.0366 | 0.0872 | 0.5583 | 0.2032 | 0.2381 |
| | 2010 | 2428 | 0.0391 | 0.0872 | 0.5895 | 0.2138 | 0.2381 |
| | 2018 | 2389 | 0.0387 | 0.0864 | 0.5882 | 0.2131 | 0.2381 |

**Table 6.** *Cont.*

| Type | Year | NP | $C_i$ | $DO_i$ | $S_i$ | $E_i$ | $F_i$ |
|---|---|---|---|---|---|---|---|
| Construction Land | 2000 | 4712 | 0.0569 | 0.1280 | 0.6163 | 0.2390 | 0.0476 |
| | 2010 | 4706 | 0.0507 | 0.1299 | 0.5490 | 0.2160 | 0.0476 |
| | 2018 | 4774 | 0.0410 | 0.1357 | 0.4411 | 0.1800 | 0.0476 |
| Unused land | 2000 | 3 | 0.0131 | 0.0004 | 5.6210 | 1.6929 | 0.2857 |
| | 2010 | 5 | 0.0259 | 0.0004 | 8.6106 | 2.5962 | 0.2857 |
| | 2018 | 8 | 0.0215 | 0.0008 | 5.6351 | 1.7014 | 0.2857 |

Note: *NP* denotes the number of patches; $C_i$ represents the landscape fragmentation index; $DO_i$ indicates the landscape dominance index; $S_i$ denotes the landscape separateness index; $E_i$ stands for the landscape disturbance index; and $F_i$ denotes the landscape vulnerability index. The calculation formula of each landscape pattern index and its ecological significance can be found in [49].

*4.3. Ecological Risk Prediction and Control*

The rapid development of urbanization significantly affects the spatial distribution of regional land, and different LU types convert into each other, resulting in changes in ecological risks. In 2036, the spatial distribution and changing trends in ecological risks in Nanning differed under various scenarios. Compared with the ecological risks in 2018, the area of high-risk regions increased under the NDS, while the area of high-risk regions decreased under the EPS, indicating that the EPS was more conducive to reducing the ecological risks in Nanning. It is worth noting that in 2036, the NDS of Nanning's ecological risk level improvement and deterioration is greater than the ecological protection area. This is because the NDS follows the law of LU change and does not impose artificial control factors and various LU type transformations, so the areas of improvement and deterioration fluctuate considerably.

Based on the current situation of the ecological environment of Nanning, combined with the ERA and the prediction results, and given the high-risk and medium-high-risk areas, we suggest strictly limiting the expansion of urban construction land disorder, focusing on the improvement of black and odorous water bodies and ecological restoration in the Youjiang, Yongjiang, Yujiang and other river basins, establishing a cultivated land protection system, and vigorously promoting land consolidation and high-standard basic farmland construction. For medium-risk areas, we suggest rationally developing and constructing the unused land, reducing the degree of fragmentation and separation of unused land, increasing forest and grass coverage, and strengthening the stability of ecosystem structure and function. For the low-risk and low-medium-risk areas, we suggest continuing to conserve forest resources such as Qingxiu Mountain and Daming Mountain and building an ecological barrier in the northern region of Nanning. In general, ecological risk aversion and ecological environment governance in Nanning should be controlled according to the regional characteristics. In addition, we suggest delineating the ecological protection red line and reasonably planning urban space, agricultural space, and ecological space.

*4.4. Study Shortcomings and Recommended Process Improvements*

Due to the complex dynamic characteristics and spatial heterogeneity in ecological processes, any scale will be affected by the interaction of social, economic, or decision-making factors at other scales, so there are often many uncertainties in the scale of ecological risks [54]. In the ERA, the division of risk plots reduces the scale effect to a certain extent, but the risk plots segment the spatial continuity of the original natural landscape. Therefore, future research can explore the spatial and temporal variation characteristics of ecological risks at multiple scales. Secondly, when the PLUS model was used in this study to simulate and predict ecological risks, no restricted conversion areas were added due to the data acquisition situation. Therefore, in subsequent studies, natural reserves, scenic spots, and restricted development zones in Nanning should be set aside as restricted conversion areas. At the same time, in the simulation and prediction of future ecological risks, only the



multi-scale effects of LU were taken into account without considering the scale effects of other driving factors, which will inevitably affect the results of ecological risk prediction. Therefore, future studies must take into account the uncertainties introduced by the drivers of varying scales when interpreting the research results. In addition, the current research on ecological risk optimization and management is relatively weak, and a scientific and mature framework system has not yet been formed.

## 5. Conclusions

Based on 30 m LU data, this study selected the best scale of ERA in Nanning from the perspective of spatial granularity and extent. It utilized the LU transfer matrix and ERA model to analyze the spatio-temporal variation characteristics of LU and ecological risk in Nanning under the best scale. At the same time, the PLUS model was used to simulate and predict the ecological risks of Nanning in 2036 under multiple scenarios and put forward ecological protection opinions, providing a reference for evaluating, avoiding, and predicting ecological risks in landscape garden cities. The conclusions are as follows:

(1)  SPLIT, SHAPE_MN, and CONTAG indices were sensitive to spatial granularity changes. The spatial granularity of 120 m was the inflection point of each landscape pattern's grain size response curve. When the spatial extent was 7 km, the semi-variance function fitting model of landscape ecological risk had the best effect. Therefore, the spatial granularity of 120 m and the spatial extent of 7 km make the best scale for ERA in Nanning.

(2)  From 2000 to 2018, the area of construction land in Nanning increased significantly, mainly from arable land and woodland. The high-risk areas were mainly distributed on unused land, construction land, and waters. From 2000 to 2018, the ecological risk of Nanning was improved, and the improvement areas were mainly concentrated in the seven districts under the jurisdiction of Nanning.

(3)  The temporal and spatial variation characteristics of ecological risk in Nanning under different scenarios in 2036 are different. The overall ecological risk in the EPS was lower than that in the NDS, and the high-risk region's area was 153.54 km$^2$ less than that of the NDS.

Our research indicates that the variation in spatial scale will affect the accuracy of landscape ERA. There are evident spatial differences in ecological risks in different regions, and targeted protection should be carried out for regions with different ecological risk levels. In addition, our study provides a framework for ERA and prediction in developing countries and landscape cities.

**Author Contributions:** Conceptualization, Y.Y.; methodology, Y.Y. and J.C.; validation, J.C. and R.H.; formal analysis, Y.Y.; resources, J.C., G.Z., H.Y. and X.H.; data curation, Y.Y. and Z.F.; writing—original draft preparation, Y.Y.; writing—review and editing, Y.Y. and J.C.; visualization, Y.Y.; supervision, J.C., R.H., Z.F., G.Z., H.Y. and X.H.; funding acquisition, J.C., G.Z., H.Y. and X.H. All authors have read and agreed to the published version of the manuscript.

**Funding:** This study was supported by grants from the Guangxi Science and Technology Base and Talent Project (GuikeAD19245032, GuikeAD19110064), the National Natural Science Foundation of China (41901370, 41961065), Guangxi Key Laboratory of Spatial Information and Geomatics under Grant 19-050-11-22, and the BaGuiScholars program of the provincial government of Guangxi (Guoqing Zhou).

**Data Availability Statement:** Not applicable.

**Acknowledgments:** The authors sincerely thank the editors and the anonymous reviewers for their constructive feedback.

**Conflicts of Interest:** The authors declare no conflict of interest.

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
