# Peer review of "Ecological Risk Assessment and Prediction Based on Scale Optimization—A Case Study of Nanning, a Landscape Garden City in China"

_remotesensing, doi:10.3390/rs15051304_

Round 1

Reviewer 1 Report (New Reviewer)

The manuscript needs to be improved in order to go ahead in the pubblication phase. The paper is quite interesting anyway some chages have to be done. If authors well follow all suggestion given I will certianly recommed this paper for pubblication on Remote Sensing

In particular, I sugget you to improve the introduction better depicting Ecological risk assessment by using Remote Sensing data in different case and also with diseases and remote sensing. To do this I strongly advice you to consider and include these papers:

- https://doi.org/10.3390/rs12213542

- https://doi.org/10.1016/j.scitotenv.2017.11.034

- https://doi.org/10.1016/j.envsci.2016.11.009

- https://doi.org/10.3390/ani12081049

- https://doi.org/10.1007/s10980-021-01346-z

- https://doi.org/10.1046/j.1365-2427.2002.00918.x

Moreover, the figure have a very low quality please improve all of them and consider to include EPSG, reference system, representation scale, nominal scale per each map.

Please consider to describe the software and the settings adopted to perfrom each analysis.

Land use image has no caption section and the % are not well describe the image has to be improved and better explained.

Author Response

请参阅附件。

Reviewer 2 Report (New Reviewer)

This study determined the optimal scale of ecological risk assessment in Nanning from spatial granularity and spatial extent based on land use data, and used PLUS model to simulate and predict the future ecological risk situation. In general, this study analyzes and predicts the temporal and spatial evolution characteristics of ecological risks, which has certain reference value for ecological protection in Nanning City.

 The purpose of the research is clear, the overall structure of the manuscript is easy to understand, and the different sections are well balanced in length and content. This manuscript presents an interesting method to study the relevant content of urban ecological risk. In addition, the technical, structural and formal quality of the manuscript is excellent, and the results are well presented and discussed with the relevant literature.

 In spite of this, the manuscript still has the following problems that need to be improved.

 1. line20 writes "in 2036 under multiple scenarios", but this study only involves two scenarios, so it is suggested to modify the expression.

 2. The research framework is the logical arrangement of the whole manuscript, which helps readers to fully understand the research content. The author's description of the research framework is too simple, please supplement the specific content according to the flow chart.

 3. What is the basis for the author to divide the five ecological risk levels in Section 2.3.5?

 4. When the author simulated the ecological risk of Nanning in 2036, how to explain that the ecological risk improvement area under the natural development scenario is greater than the ecological protection scenario?

 5. In the conclusion part, it is necessary to provide specific public policy suggestions from the research results to provide reference basis for policy recipients and decision makers.

Author Response

Reviewer 3 Report (New Reviewer)

  Ecological risk evaluation of Nanning city was conducted in this manuscript, and substantial results were concluded. However, the methods were not innovative, and many technical details were not adequately explained. Some universal issues were proposed in discussion section, and   key problems and solutions for this manuscript were not involved. Moreover, English vocabulary is cryptic and challenging to understand. Overall, it is not suggested to publish in Remote sensing.

1. Line 188: Why the sizes of grid ERA were set as 4*4km2,5*5km2....Are there any foundations to support this?

2. Line 192: How did the authors selecting the 17 landscape indexes? Are the 17 landscape indexes representative? It is crucial for the decision of the threshold.

3. Line 249-252: Please clarify how was the Ri calculated in Equation (4).

4. The index orders in figure 4 were different with those in figure 5. To ensure the readable of the figures, please keep the indexes sorted.

5. Line 283-292: The scenario setting was not representative and connected with local policy initiatives. To distinguish the scenario from other studies and serve as the foundations for specific suggestions, please express the unique characteristics of the scenario setting.

6. Line 325: As seen in figure 5, 60m mark was also the inflection point of the fold lines. Why did the authors select 120m mark? Clarify the theoretical bases.

7. Line 405: Which year was calculated in figure 8? Mark the years in the figure or in the words.

8. Line 500-511: Section 4.2 failed to clarify the improvement of landscape measures and suggests. Those measures such as black smelly water remediation are the main reason for the ecological risk reduction in the study area. The landscape indicators were used for ecological risk assessment in this manuscript, which did not explain well why the ecological risk is reduced.

9. The proposed measures in section 4.3 were conventional and could not reflect the characteristics of the study area and the differences with other regions.

Round 2

Reviewer 1 Report (New Reviewer)

The authors well follow the suggestion given. I only recommend to include the references mentioned below when you tell in the manuscript in line 68-70 "Although current studies have explored the effects of suitable spatial grain size on landscape pattern changes from a landscape pattern perspective, the impact of scale effects on landscape ecological risk has been ignored." 

Please add these references to justify the statement:

- https://doi.org/10.3390/rs15010178 

- https://doi.org/10.3390/app13010390

- https://doi.org/10.3390/land10121368

Then, according to the present reviewer the manuscript can be published on Remote Sensing now the quality very is high.

Author Response

Reviewer 3 Report (New Reviewer)

1. Innovation is still insufficient in this  article. Both spatial scale and ecological risk are popular and mature topics in Ecology, so spatial scale cannot be considered as technical or applying innovation.

2. I cannot agree with the author's reply in Point 7, about the selection of the best landscape granularity. As in figure 5, AREA_MN、NP、PD、SPLIT also have inflection point at 150m, why did not the authors consider 150 m as the inflection point? Theoretically, the large scale in the first scale domain should be selected as the best scale; meanwhile, the authors should test  the variations of different spatial scales to prove the best one using related index (e.g. entropy). It is necessary to be rigorous in this step, which is directly related to the accuracy of final evaluation.

Overall, this paper is not applicable for RS.

Author Response

This manuscript is a resubmission of an earlier submission. The following is a list of the peer review reports and author responses from that submission.

Round 1

Reviewer 1 Report

I found the manuscript to be interesting and has a good potential in the ecological risk assessment and prediction based on scale optimization. This study investigated the capability of analysis and prediction of urban ecological risk using 30m land use data and exploring the patterns of spatial and temporal changes in ecological risk in Nanning on the optimal scale. The results show that a spatial granularity of 120 m and a spatial extent of 7 km are the best scales for ERA and prediction in Nanning. The results can provide theoretical support for ERA and prediction of landscape cities and ecological civilization construction. I find that the introduction provides decent background, the research design is appropriate, the methods adequately described, and results are presented, so I do not have many major comments, but have two general question about the uncertainty of data and models. I believe a major level of revisions should be made to the paper before it is ready to be considered for publication with Remote Sensing.

General:

Authors using multiple sources of data in different spatial resolutions, what is the uncertainty of data?

What is the major uncertainty of the models?

Reviewer 2 Report

This paper is not related to remote sensing at all. No remote sensing images are used here. It is possible that the land use data product at 30m resolution, from the Chinese Academy of Sciences, may have originally been derived from remote sensing but even this infomation is missing. The paper is therefore not suitable for publication in Remote Sensing. Journals in the fields of GIS, Urban Planning, Landscape design etc. would be more suitable for this research.